MINIREVIEW
# Methods and Strategies to Uncover Coral-Associated Microbial Dark Matter

Júnia Schultz,[a] Flúvio Modolon,[b] Alexandre S. Rosado,[a] Christian R. Voolstra,[c] Michael Sweet,[d] Raquel S. Peixoto[a]

[a]Red Sea Research Center, King Abdullah University of Science and Technology, Thuwal, Saudi Arabia
[b]Laboratory of Molecular Microbial Ecology, Institute of Microbiology, Federal University of Rio de Janeiro, Rio de Janeiro, Brazil
[c]Department of Biology, University of Konstanz, Konstanz, Germany
[d]Aquatic Research Facility, Environmental Sustainability Research Centre, University of Derby, Derby, UK

Júnia Schultz and Flúvio Modolon contributed equally to this article. Author order was determined as Júnia has started the writting.

**ABSTRACT** The vast majority of environmental microbes have not yet been cultured, and most of the knowledge on coral-associated microbes (CAMs) has been generated from amplicon sequencing and metagenomes. However, exploring cultured CAMs is key for a detailed and comprehensive characterization of the roles of these microbes in shaping coral health and, ultimately, for their biotechnological use as, for example, coral probiotics and other natural products. Here, the strategies and technologies that have been used to access cultured CAMs are presented, while advantages and disadvantages associated with each of these strategies are discussed. We highlight the existing gaps and potential improvements in culture-dependent methodologies, indicating several possible alternatives (including culturomics and *in situ* diffusion devices) that could be applied to retrieve the CAM "dark matter" (i.e., the currently undescribed CAMs). This study provides the most comprehensive synthesis of the methodologies used to recover the cultured coral microbiome to date and draws suggestions for the development of the next generation of CAM culturomics.

**KEYWORDS** coral microbiome, coral metaorganism, culturomics, culture-dependent techniques, microbial dark matter, coral-associated microbes, coral probiotics, culturing

**B**eneficial microbes are essential members of the coral holobiont (i.e., the coral host plus the associated microbiome, including their endosymbionts) (1, 2) as they can contribute to energy and nutrient input, remediation of toxic compounds, and mitigation of environmental and/or pathogenic impacts (3, 4). Exploring the diversity of cultured coral-associated microbes (CAMs), both in terms of taxonomy and functionality, is therefore an important step to fully characterize and elucidate the roles these microbes play in shaping host health. Such baseline information is also useful for biotechnological and experimental applications, such as the use of coral probiotics (5–11).

The vast majority of microbes are still unknown. Lagier et al. (12), for example, were only able to assign a species ID to 51 of the 340 cultured species from the human microbiome. Most of the knowledge of CAMs has been generated from amplicon sequencing based on PCR amplification of specific gene markers (16S, 18S, and 23S rDNA) (2, 13) and/or more recently through metagenomics (14–16). Such tools have generated important knowledge regarding coral-microbiome assemblages and environmental interactions (17), microbial transmission (18), and coral holobiont plasticity and physiology (19), as well as the microbiome response to environmental impacts (9, 10, 16, 20–24).

Multi-omics analyses have also revealed the extent of the uncharacterized and uncultured majority of CAMs (14, 21, 25, 26), which can be orders of magnitude higher than those culturable within laboratory settings (27, 28). In addition, molecular tools may inform culturing strategies on how to improve the culturability of the "hidden

Address correspondence to Raquel S. Peixoto, raquel.peixoto@kaust.edu.sa.

The authors declare no conflict of interest.

diversity" that appears to be out of reach (29, 30). However, these same tools might be insufficient to confirm specific functions, understand the physiology of unknown taxa, and explore mechanisms associated with the complex interactions between hosts and microbes, which highlights the importance of cultivation in the age of multi-omics.

The first big effort to pull together the current state of play of cultured CAMs was recently performed by Sweet and colleagues (28). The authors analyzed a total of 3,055 prokaryotic isolates and 74 genomes from 52 studies (published and unpublished). Through cultivation, 138 bacterial genera were recovered and the most abundant genera isolated were *Ruegeria*, *Photobacterium*, *Pseudomonas*, *Pseudoalteromonas*, *Vibrio*, *Pseudovibrio*, and *Alteromonas*; the majority of these were cultured from marine agar (715 distinct isolates). Furthermore, only a very small fraction of the isolated strains also had their genome sequenced. Another study by Huggett and Apprill (27), taking a slightly different approach, highlighted similar trends; only 6.5% of the 21,100 sequences available in the Coral Microbiome Database (https://vamps2.mbl.edu/portals/CMP) were generated from culture isolates. Indeed, 87.4% were identified as uncultured (PCR-based methods or clone libraries) (27). Additionally, only 14 of the 41 taxonomic groups assigned contained cultured representatives (27), which is also aligned with the fact that less than 50% of the genes from the so-called "microbial dark matter" (i.e., the yet-to-be-cultured microbes) can be annotated (31).

This lack of cultured representatives for many of the "known" bacterial groups reinforces the need for more efficient tools to culture these organisms, especially considering many act to effectively repair or restore the coral microbiome to promote health and fitness (3, 4, 13, 32, 33). Several strategies that have been used in other systems and organisms (e.g., soil and humans) (34, 35) could now be explored to spur new developments in coral studies. For example, innovative culture media and modifications in the incubation parameters could be employed, as well as alternative tools such as diffusion-based devices for *in situ* cultivation (36–38) and cell-targeted cultivation (39, 40).

This review aims to provide an overview of the strategies and technologies that have been currently used to access cultured CAMs and highlights the advantages and disadvantages associated with these strategies, as well as the existing gaps and improvements in culture-dependent methodologies. We then propose several possible alternatives that could be applied to retrieve the yet-to-be-cultured members of the coral microbiome and thereby shed light on their potential application in ecology and biotechnology. This study provides the most comprehensive synthesis of the methodologies used to recover the cultured coral microbiome to date.

## ACCESSING THE CULTURED CORAL MICROBIOME

Culture-based tools are mainly focused on standardized methods, using nutrient-rich or nutrient-poor culture media and standard incubation time, pH, salinity, oxygen, and temperature (28, 41). The use of alternative approaches, such as different cultivation settings (41, 42) and *in situ* cultivation using diffusion-based devices, for example (43, 44), could overcome challenges in mimicking the natural environment and isolating "hard-to-culture" microbes (e.g., *Endozoicomonas*, see Pogoreutz and Voolstra [45]). In fact, both conventional methods and alternative culture approaches only enable the retrieval of a small fraction of the real microbial diversity and cell abundance in any environment, due to the phenomenon termed "the great plate count anomaly" (46), which therefore highlights the necessity to apply combined tools to expand our reach.

Although there is no consensus on the exact percentage of currently cultured bacteria and archaea, most taxa are still thought to be uncultured under laboratory conditions (reviewed in Martiny [47] and Steen et al. [48]). Moreover, a staggering 88% of those that have been cultured are from three phyla (Firmicutes, Actinobacteria, and Bacteroidetes) (49). The remaining taxa can therefore be classed as "microbial dark matter." Here, we discuss most techniques and strategies adopted to overcome such "microbial unculturability." In addition to the difficulty of mimicking natural-growth conditions, sample processing can also negatively affect the viability and, consequently, the culturability of microbial cells from

samples (50), and ways to detach microbial cells from coral mucus, tissues, and skeletons must also be considered.

**Sample processing interferes with the culturability of the coral microbiome.** The first step to successfully access the culturable fraction of the CAM is to effectively detach the microbial cells from the host. If the specific location of the isolate within the holobiont is not relevant, no advanced skills and/or tools are necessary; whole coral maceration (51) is simply followed by serial dilutions and inoculation on the culture media. However, if the goal is to retrieve microbes associated with specific parts of the coral (i.e., mucus, skeleton, or tissue), specific protocols are required.

Microbes associated with mucus, for example, can be collected using sterile syringes to extract the liquid from the coral surfaces or even from the whitish slurries that can form around the corals (52, 53). Alternatively, corals can be placed upside down in sterile flasks without water, from which mucus secretion can be retrieved after 20 to 30 min, sometimes referred to as coral "milking" (54). Also, a commonly used technique for sampling mucus is swabbing the coral surface (55, 56). Additionally, the mucus can be sampled by low-speed centrifugation (57, 58). Finally, the aptly named "snot sucker," an apparatus that allows for the separation of the two distinct layers of the surface mucus, can also be used (59). Each of these protocols present associated challenges and limitations, including the amount of biomass that is obtained, potential microbial contamination from the surrounding seawater, and/or cross-contamination among coral compartments. The selected technique will therefore likely rely on the available expertise, logistics, and research goals.

For coral tissue sampling, external layers of tissue, such as the epidermis and gastrodermis, can be detached from the skeleton by airbrushing with sterile buffers or filtered seawater (59, 60). The yield recovered can vary depending on the applied pressure and rigidity of the bristle material. Calcium- and magnesium-free seawater incubation has also been used due to the spontaneous detachment of coral tissue from the underlying skeleton. Intracellular microbes can be extruded by mechanical rupture of the host cells using different strategies (61), such as sonication or vortexing with or without beads. Importantly, severe mechanical disruptions must be avoided as they can damage microbial cells to the extent that they are unviable for culturing. It is important to mention the potential effect of coral holobiont compounds on the culturability of specific CAMs. The production of mantiporic acids by the stony coral *Montipora* spp. (62, 63) can, for example, affect cultured cells (e.g., zooxanthellae), due to its antimicrobial activity and cytotoxicity (63, 64).

The symbiosome and associated Symbiodiniaceae (65) also lies within the tissue; accessing the CAMs associated with these would require successive steps of tissue extraction and washing, followed by filtration and fractionation using a Percoll gradient as described by Peng et al. (167). This approach, applied for proteomic analysis, maintains the integrity of symbiosomes and associated Symbiodiniaceae. Overall, because sample processing interferes with the culturability of CAMs, methods inducing minimum cell damage should be employed.

Interestingly, and somewhat surprisingly, the majority of CAMs are in the skeleton, living in porous forms in different extracts (59, 66). Dissolution of the skeleton by decalcification can be performed to access the entire genomic content of skeleton-associated microorganisms (SAMs). However, to maintain the viability of microbial cells for culturing, the skeleton can be crushed with a mortar and pestle and shaken overnight with glass beads. Using this method, small fragments typically remain intact, which means it is unlikely that all the SAMs present will be detected. Furthermore, the coral skeleton presents a gradient of pH, oxygen, light, and nutrients, among other parameters, which forms different submicroenvironments that are occupied by their own unique microbes (66), and many of these would not be extruded using this method alone.

**Culture-based approaches.** In this section, we examine the tools used for the cultivation of microorganisms and highlight alternative approaches to improve CAM culturability (Table 1).

**TABLE 1** Summary of approaches and strategies to improve microbial culturability

| Approach | Technique | Focus | Description | Previous uses | Advantages | Disadvantages | Year of publication/reference |
|---|---|---|---|---|---|---|---|
| Culture media and incubation improvement | Commonly used media with or without modifications to content and incubation conditions | Generalist and specialized microbes | Use of regular culture media with or without modifications to pH, composition, air conditions, incubation time, inoculum size, temp; aiming to mimic natural conditions | Environmental and clinical microbes | Easy formulation; Easy manipulation; Routine methods and commercial options; Low cost; No robust equipment or techniques required | Cultivation bias toward four Phyla: Proteobacteria, Firmicutes, Bacteroidetes and Actinobacteria | Esteves et al. (28), Pulschen et al. (41), Rygaard et al. (50), Sweet et al. (71), Zheng et al. (75) |
| | Coculture | Autotrophic and syntrophic microbes | Cultivation of "not-yet-cultured" microbes that require other microbe(s) to grow, and/or assistance of helper microbe(s) | Environmental and clinical microbes | Low cost; Growth of microbes that depend on specific (and eventually unknown) substances produced by other microbes | May require optimization due to different nutrient requirements; Lack of pure cultures | D'Onofrio et al. (90), Marmann et al. (96) |
| | Culturomics | Generalist and selective microbes | Use of an array of culture media with difference compositions and incubation conditions to determine the best approach | Human gut microbiome, environmental microbes | High-throughput isolation; Simultaneous microbial growth under different conditions | Large amounts of samples to process | Lagier et al. (12), Lagier et al. (42) |
| Diffusion-based devices and *in situ* cultivation | Cultivation chambers | Generalist microbes | Device with single compartment for *in situ* cultivation, based on microbial growth by diffusion of growth factors from the environment. It can hold many cells in the single compartment | Environmental microbes | *In situ* cultivation | Competition between the cells inside the chamber can occur, which drives selectivity | Gavrish et al. (43), Steinert et al. (103) |
| | Isolation chips | Generalist microbes | Device with several wells for *in situ* cultivation, based on microbial growth by diffusion of growth factors from the environment. Single or few cells per well | Environmental microbes | High-throughput cultivation; *In situ* cultivation | Difficult to load cells into wells | Berdy et al. (44) |
| | Diffusion bioreactor | Free-living microbes | Use of a reactor for microbial growth using diffusion of growth factors, mimicking natural environment | Soil microbes | High-throughput cultivation; Mimics natural conditions | Needs a solid matrix for the exchange of growth factors with culture media | Chaudhary et al. (36) |
| | Sphere of the gelatin agent | Free-living microbes | Spheres containing entrapped microbes that are coated with polymer for *in situ* incubation | Environmental microbes | Replaces the use of supports made of potentially toxic materials | Competition between the cells inside the chamber can occur, which drives selectivity | Kushmaro and Geresh (109) |
| | Microbe Domestication Pod (MD Pod) | Generalist microbes | Microcapsules of agarose loaded with single bacteria, bounded by chambers (Pod) | Marine sediment microbes | High-throughput cultivation; *In situ* cultivation | Difficult to sort and load cells into microcapsules | Alkayyali et al. (111) |
| | Multiwell microbial culture chip | Generalist microbes | Micro-petri dish with millions of compartments to grow different cultures | Freshwater | High-efficiency cultivation; High-throughput cultivation; Allows screening for specific phenotypes | Difficult to pick/recover microcolonies | Ingham et al. (107) |
| | Hollow-fiber membrane-based | Free-living microbes | Microbial growth by diffusion of growth factors from the environment, mimicking the natural environment. Devices constituted by porous flexible pipes with injectors | Marine microbes | Injectors maintain the flow of substrates inside the device, improving the culturability | Oversize; Culture of microbes from a thin and more superficial water layer | Aoi et al. (112) |

**TABLE 1** (Continued)

| Approach | Technique | Focus | Description | Previous uses | Advantages | Disadvantages | Year of publication/ reference |
|---|---|---|---|---|---|---|---|
| | I-Tips | Host-associated microbes | Device developed for *in situ* cultivation, based on microbial growth by diffusion of growth factors from the environment. Microbial cells can transit from the outside to the inside of the device | Host-associated microbes | Low cost | Without growth control | Jung et al. (38) |
| | Paper-based analytical device (PAD) | Generalist microbes | *In situ* cultivation | Human gut microbes | Multifunctional Low cost Efficient for clinical diagnostics | Selective growth of well-known cultured microbes | Noiphung and Laiwattanapais et al. (114) |
| Targeted culturing | Reverse Genomics | Targeted microbes | Capture and cultivation of targeted microbes using genome-informed antibody approaches | Human oral Saccharibacteria | Isolation and cultivation of targeted microbes for specific studies | Expensive | Cross et al. (39) |
| | Live-FISH | Targeted microbes | Capture of targeted microbes using living cells labeled with DNA probes and cell sorting and followed by cultivation methods | Marine microbes | Isolation and cultivation of targeted microbes to specific studies | Variations in fluorescent signals can cause difficult detectability of some microbes | Batani et al. (35) |
| | Micromanipulators and laser manipulation system | Generalist and selective microbes | Isolation of targeted single cells from a mixed community based on trapping microbes to be separated through microscopy, optical tweezers or laser energy | Marine microbes and human cells | Selection of cells of interest from a mixed microbial community Pure colonies obtained by sorting cells with microscope | Technical abilities required Laser manipulation | Fröhlich et al. (40), Zhang et al. (117), Keloth et al. (118), Lee et al. (119) |
| Other | Winogradsky column | Generalist microbes | Container loaded with solid matrix and water, containing different microhabitats with different grades of oxygen and carbon sources | Environmental microbes | Easy to prepare Low cost Different oxygen demand conditions | Difficult to capture and isolate microbes | Gutleben et al. (116) |

**(i) Culture media and growth conditions: regular and modified media.** Many resources are required for a microbe to be cultured, such as the correct physicochemical nutrients and energy sources. The most basic approach used to culture microbes from different sources is the use of nutrient-rich culture media (e.g., tryptic soy broth, lysogeny broth, marine broth, nutrient broth, to name a few). These culture media were developed decades ago (more than 50 years) and have been applied to grow and maintain microbial cultures from different environments and hosts, including the coral holobiont, using relatively few modifications that can increase the microbial diversity obtained. Here, we highlight that novel approaches developed for other hosts and samples should be adapted to increase CAM culturabilty.

Sweet et al. (28) have recently summarized the available data on the cultured fraction of the coral microbiome. The authors observed that the most common culture media used by coral microbiologists are marine agar (MA) and glycerol artificial seawater agar (GASW Agar), of which, according to the applied methods, coral samples, and results obtained, MA has been proven to be the best culture medium to retrieve a high number of unique isolates (28). Based on this study (28), the general culture media nutrient agar seems to recover the higher diversity of CAMs, followed by R2A. Conversely, thiosulfate-citrate-bole-salts-sucrose agar, nitrogen-free medium (NFb 256), and GASW Agar seem to provide the lowest bacterial diversity of current cultured coral-associated bacteria (28).

In addition, the selective DMSP-enriched culture medium seems to be a good option to increase the diversity of obtained CAMs (28), which is an example that, despite the potential accumulation of selective by-products from the metabolism of culture media additives, such as sugars and or chemical compounds like DMSP (67), the selection of the media composition should be based on the scientific goals.

General culture media enable the growth of a higher abundance of microbes, albeit with low overall diversity (50). In contrast, selective media enable the growth of specific bacteria that would not be necessarily isolated by general media. Thus, combining different culture media and applying variable incubation conditions will undoubtedly increase the diversity of cultured bacteria. For example, Esteves et al. (50) used this approach and obtained 15% of the total bacterial diversity from a sponge.

Media customization is an alternative approach that may enhance the diversity of isolated microorganisms, such as modifications to media composition that may include the addition of specific chemicals, the use of specific carbon sources, and use of alternative gelling agents. Such customizations benefit from in-depth insight into the nature and characterization of samples, to better understand the factors involved in their growth under natural conditions and potential improvements to retrieve specific microbial groups (68).

The excessive amount of nutrients from rich culture media can cause a high-nutrient shock and impair the growth of microorganisms from stressful low-nutrient environments (69). Keller-Costa et al. (70) isolated a plethora of strains associated with the soft coral *Eunicella labiata* using a basic modification in marine agar, which was diluted (1:2) in artificial seawater, aiming to deplete part of the carbon source. With this relatively simple modification, 416 morphologically distinct colonies were obtained, corresponding to 62% of the bacterial phylotypes associated with the gorgonian. Another example of a relatively simple, although effective, modification is the culturing of the first free-living representative of *Candidatus Izemoplasma* from deep-sea cold seeps achieved with the addition of an enrichment step to the conventional culturing approaches (71). In brief, sediment samples were enriched for 6 months using an anaerobic basal medium supplemented with 1.0 mg/L *Escherichia coli* genomic DNA at 28°C, as previously described by Fardeau et al. (72). The extracellular DNA was then used as a source of phosphorus, carbon, and nitrogen to improve the culturability of this deep-sea species.

The development of new culture media recipes that mimic the environment from which samples are obtained has also contributed to the ability to grow previously

uncultured bacteria. For example, the use of soil-extract agar medium (73) has enabled new species of bacteria to grow due to the presence of soil constituents that are required for the growth of specific groups of microorganisms (e.g., Actinomycetes) (73). In a similar approach, Olson et al. (74) tested different conditions to improve the culturability of microbial colonies from marine sponges and observed an increase in microbial CFU when sponge extract was added to the regular culture media. Following the same principle, we propose the use of a coral juice medium based on macerated coral as a substrate that could, theoretically, allow the growth of CAMs that rely on compounds present in the host (see detailed information in Future Prospects).

Another important and overlooked factor that can affect the culturability of microbes is the gelling agent. This is a key driver of the structure of different cultured communities (75). Agar, a product derived from a group of red-purple marine algae (e.g., the genera *Gelidium* and *Gracilaria*), was introduced for microbiology in 1882 and continues to have a major impact, being the most used gelling agent (76). However, agar may also have negative effects on microbial culturability, such as the formation of reactive oxygen species during sterilization of the agar. To overcome this limitation, similar gelling properties can be found in various other agents, such as xanthan gum, carrageenan, isubgol, gellan gum, and acetan, an exopolysaccharide produced and secreted by the aerobic Gram-negative bacterium *Acetobacter xylinum*. These polymers can be used, for example, to culture organisms that do not grow or grow poorly on agar and may increase the number of CAMs and possibly even their respective growth rates (77, 78).

Gellan gum, for example, is an extracellular polysaccharide secreted by the bacterium *Sphingomonas elodea* (79) that is commercially manufactured by a fermentation process. The unique colloidal and gelling properties of gellan gum were discovered in 1978 and present good ability to form coatings and high clarity (77, 80), and, due to its high thermal stability, it has been often used to culture thermophiles (81, 82). Additionally, several studies have observed that gellan gum improves the culturable diversity of some microorganisms, such as rare thermophilic Actinobacteria, several anaerobic (hyper) thermophilic marine microbes, and previously uncultured bacteria from soil (82–84). Sugars from gellan gum (e.g., glucose, rhamnose, glucuronic acid) may stimulate advanced microbial growth (77).

Additionally, a "selective medium-design algorithm restricted by two constraints (SMART)" can be applied (85) and is based on two selective agents: (i) a specific carbon source, enabling the proliferation of the target microorganism, and (ii) antimicrobials, suppressing unwanted microorganisms. Described in 2011, this method was again used to facilitate the development of selective media targeting key specific soil microbiota, such as *Burkholderia glumae*, *Acidovorax avenae*, *Pectobacterium carotovorum*, *Ralstonia solanacearum*, and *Xanthomonas campestris* (85).

Another strategy adopted by some microbiologists is changing growth conditions that are important for microbial cultivation, such as the incubation period, inoculum size, temperature, pH, and atmospheric conditions ($CO_2/O_2$ level). One such strategy is lowering the temperature and increasing the incubation period, as well as incubating plates in the absence of light or inside polyethylene bags to avoid desiccation (41, 70).

For slow-growing bacteria, different strategies of modified "culturomic" approaches can be applied to improve their recovery, especially due to the bias in cultivation by competition with fast-growing bacteria (86). To recover a greater diversity of slow-growing bacteria, the use of low-nutrient composition media (41) and supplementation with water from the sample source (87) has also been used. As long periods of incubation are required for the growth of slow-growing bacteria, the use of antibiotics (e.g., Amphotericin B) is also indicated to prevent fungal contamination. Additionally, incubation in the absence of light and at temperatures of approximately 12°C is useful (41).

Extracellular signaling molecules, such as cyclic AMP and acyl homoserine lactones (AHL), can also be successfully used as supplements to enhance the culturability of

some groups of microbes (75, 88). However, in other cases signaling molecules (i.e., AHLs) have negatively impacted the number of viable counts recorded (75, 89).

**(ii) Coculture.** Mimicking naturally occurring biological interactions can also help overcome limitations associated with traditional cultivation (90). Complex networks are commonly observed in natural environments, where microbes can cooperate through the exchange of metabolites and signaling molecules, as observed in changeling environments and biofilms (91). Several obligate symbionts are hard to culture in laboratory conditions because they have coevolved with their host or microbial partners (91, 92). A strategy to overcome this limitation is coculturing interdependent microbes.

Coculture methods are widely used in synthetic biology and for the production of bioactive secondary metabolites (93). In addition, coculturing may activate genes that are not expressed under normal laboratory conditions, thus stimulating metabolic pathways that are not active in pure cultures (94). Several studies have successfully used the coculture strategy, allowing the recovery of several previously uncultured bacteria, for example, microorganisms from marine invertebrates and algae, marine sediments (95, 96), and sponge-associated bacteria (97, 98). To the best of our knowledge, studies applying coculture on coral samples have not been undertaken to date.

**(iii) Alternative approaches.** Efforts to improve the culturability of environmental microorganisms by using cocultures, alternative culture media, and incubation strategies appear to have contributed to increase the recovery of some microbial groups. Nevertheless, most bacterial groups remain uncultured. Here we summarize innovative approaches that have been developed to overcome bacterial culturability limitations.

**(a) Culturomics.** Following some adaptations and new strategies proposed in the early 2000s, in 2012, Lagier et al. (99) initiated the "microbial culturomics" approach and paved the way for the next generation of microbial cultivation techniques. Culturomics may be defined, by analogy with metagenomics, as an approach allowing extensive assessment of microbial composition by high-throughput culture (100). This includes exploring the aforementioned improvements to culture methods/techniques (e.g., low-nutrient media, addition of host-tissue extracts and/or substrates, signaling compounds, coculture, variations in pH, temperature, light incidence, salinity, oxygen demand, etc.), and the use of high-throughput approaches (e.g., matrix-assisted laser desorption ionization–time of flight mass spectrometry and 16S rDNA sequencing) to retrieve large numbers of microbial colonies. This combined use of modified culture conditions and high-throughput approaches has allowed the recovery of 384 previously unknown microbial species (12) and the retrieval of 497 hitherto species in studies of the human gut (101).

High-throughput cultivation pipelines are also becoming available, such as the so called "Culturome" designed for the culturing of plant root-associated bacteria using limiting dilution in 96-well plates (102). The culturome effectively guarantees the growth of axenic colonies, enabling the retrieval of slow-growing bacteria, ensuring independence from those that are fast growing (102). Moreover, this approach describes an integrated workflow to facilitate the identification of colonies using marker-gene amplicon sequencing (102) and could be also adapted for marine samples.

**(b) Diffusion-based devices in microbial cultivation.** To culture marine sponge-associated bacteria, Steinert et al. (43) notably used diffusion chambers for *in situ* cultivation in 2014. The chambers were bound by two membranes that enable diffusion of growth factors present in the natural environment. Samples of diluted marine sponge tissue (*Rhabdastrella globostellata*) were loaded in the chambers and returned to the habitat. Sequencing of 16S rRNA and phylogenetic analyses showed that this approach enabled an increase of 339% in sponge-associated bacterial culturability. Moreover, 19.6% of sequences were from previously uncultured bacteria, mainly Proteobacteria and Bacteroidetes. A similar approach was applied for the cultivation of Actinobacteria (103). In this case, one of the two membranes had a pore size that selectively allows penetration of filamentous Actinobacteria, while simultaneously preventing penetration of other microorganisms. Once the devices are incubated in the soil, Actinobacteria grow into the

chambers and facilitated by the assimilation of specific growth factors naturally present in the environment (103). This trap allowed higher retrieval (200%) of Actinobacteria in comparison with traditional culture media, including rarely cultured genera. While traps loaded with agar enabled the isolation of 69 Actinobacteria from four genera (81% *Streptomyces*), those loaded with gellan gum allowed the isolation of 81 strains from 11 different genera. In comparison, Petri dishes containing agar and gellan gum inoculated with samples from the same source as the trap, allowed the recovery of only 25 and 41 Actinobacteria, respectively (103).

Isolation chips (iChips), developed by Berdy and colleagues (44) in 2017, are another example of a diffusion-based device that can increase the number of microbial isolates by up to 300-fold, while also recovering previously uncultured microbes. iChips have many wells inside from which it is possible to grow axenic colonies from loaded microbial cells. This technology has allowed the recovery of a strain of *Eleftheria terrae*, which produces an antibiotic named teixobactin (104). This antibiotic was the first new class of antimicrobials found after 30 years, representing a promising alternative chemotherapeutic agent to control infections caused by multidrug-resistant bacteria. Other studies have shown the applicability of iChip-like devices to culture microbes from aquatic environments and organisms, such as from mangroves (105) and sponges (37).

Early limitations associated with the quantification and sorting of the microbial cells loaded into iChip wells have recently been addressed by Liu et al. (106) through the development of the FACS-iChip, which is a flow cytometry-based and fluorescence-activated cell sorting (FACS) integration of a modified iChip. This method, with a single microbial cell per well, increased the culture recovery at up to 40% (106).

Another example of a diffusion-based approach was developed relatively recently by Chaudhary and colleagues in 2019 (36), who designed a diffusion bioreactor and used it to analyze soil samples. Briefly, a sample was loaded in a container wrapped with a polycarbonate membrane (0.4 $\mu$m) filled with culture media, and the chamber was surrounded by soil. Similar to other diffusion-based tools, the membrane enables exchange of growth factors and key compounds with the environment, thus allowing growth of incubated microbes.

Multiwell microbial culture chips are another microbial culturing alternative which use a ceramic plate with a million microwells, enabling the growth of separate microcolonies (98). This high-throughput system has been shown to promote the rapid growth of novel bacterial species with biotechnological applications (107). Similarly, Palma Esposito et al. (108) developed a Miniaturized Culture Chip (MCC), which consists of a microscreen plate. The authors used this device to grow microbes from Antarctic sediment samples and isolated the rare genus *Aequorivita* sp., strain 23L, able to produce substances with biotechnological value (108).

A gelatin sphere composed of culture medium and the microbial sample, coated with natural or synthetic polymer, forming a polymeric membrane was also developed for *in situ* incubation in natural environments (109). This approach avoids the use of support materials, such as metals, that can negatively affect the environment and inhibit cell growth. Similarly, Zengler et al. (110) loaded single cells into microcapsules that were exposed to a continuous flow of culture media under *in vitro* conditions, resulting in the recovery of approximately 10,000 microcolonies of environmental microbes. Recently, Alkayyali et al. (111) designed a platform called a Microbe Domestication Pod (MD Pod). The MD Pod has been used to load agarose microbeads containing marine bacteria and prepared by microfluidics, which serve as substrate for single-cell growth. Pod devices are chambers bound and sealed by membranes that allow the passage of growth factors from the natural environment. Once assembled, MD Pods can be incubated *in situ* (marine sediments) (111).

Other examples include an *in situ* hollow-fiber membrane platform coupled with injectors to maintain flow of substrates, which has been shown to improve the culturability of inoculated cells from aquatic environments by up to 12.3%, in contrast to 2.1% of the microbial cells inoculated in Petri dishes (112). The use of transwell plates

to allow the diffusion of nutrients from the environment is another method which, for example, has been used for the recovery of methane-oxidizing bacteria (113). Finally, the so-called "I-tip," a simple adaptation using a 200-$\mu$L pipette tip and glass beads (50 to 212 $\mu$m in diameter), loaded with culture medium R2A, has been used to enhance the growth of sponge-associated bacteria (38). All the diffusion-based alternatives discussed above could be adapted and employed for the recovery of CAMs, both in aquaria or the natural environment, by attaching these devices around or inside coral colonies.

**(c) Paper-based analytical devices.** A simple paper-based analytical device (PAD) for *in situ* cultivation and screening of *Escherichia coli* human infections was designed using wax and Whatman No. 1 filter paper combined with a cotton sheet, with the addition of agar to allow the growth of microbial cells (114). This platform also enables testing for the presence of nitrite, which indicates *E. coli* infection. The principle of this device may be applied to other microorganisms and hosts, including those from the marine environment, where PAD may be adapted to identify coral infection with *Vibrio coralliilyticus* or other pathogens using, for example, selective media and biochemical traits.

**(d) Winogradsky columns.** Winogradsky columns consist of transparent cylinders filled with particles forming a solid matrix (e.g., sediment or soil with microorganisms of interest, or even sterile silicate sand), plus a liquid phase (115, 116). These columns are commonly incubated under light regimes, and these conditions confer an ideal mesocosm for enrichment cultures (115). Biotic and abiotic processes inside the columns drive the formation of a plethora of stratified microhabitats, forming a chemical gradient suitable for different microbial-growth demands (115, 116). Recently, in 2020, this tool has been employed to improve the culture of sponge-associated bacteria (116). The authors observed that the oxygen gradient inside the columns enriched different taxa according to specific oxygen demands. Moreover, other cultivation conditions were tested including a simple dilution of culture medium up to 50 times, which notably improved microbial growth. A similar stratified matrix using components of coral skeleton, specific mucus-enriched polysaccharides from different coral species, tissue, and seawater could be used to culture CAMs.

**(e) Cell-targeted cultivation.** Many strategies have been developed for host-associated microbial-cell sorting, although they have never been applied to explore CAMs. "Reverse Genomics," for example, is a method that uses specific antibodies previously known to be associated with certain microorganisms, to select and culture a target microbe (39). Purified antibodies are loaded into the host to target a specific strain and isolate it from complex microbial communities for culturing and genome sequencing. Cross et al. (39) applied this method to isolate and culture three previously uncultured Saccharibacteria (TM7) lineages from the human oral cavity. Similarly, Live-FISH, a methodology that couples a modified pipeline of fluorescence *in situ* hybridization (FISH)-labeled DNA with probes and FACS, allows the recovery of target microbes from mock and natural communities (35).

Several methodologies using micromanipulators or laser manipulation systems (e.g., optical tweezers and laser microdissection) have now also been developed with the aim of sorting, culturing, and isolating single bacterial cells. These techniques utilize microscopy to visualize, trap, collect, and transfer a single cell out of a mixed microbial community (117–119). More recently, Lee et al. (40) proposed a similar technique labeling living microbes using stable-isotopes to select targeted cells by Raman microspectroscopy. This method allows cell sorting by functional traits before isolation or single-cell genome sequencing. Microbial-targeted culturing techniques using genome-informed probes that have been developed for other hosts and environments represent alternatives to sort and isolate specific and uncultured microbes from the coral microbiome. Although choosing the cultivation approach will depend on several variables, including the goals and available infrastructure and material, the use of multiple strategies can increase the diversity of microbes recovered from marine samples by up to 45% (120) and is therefore highly recommended when a wider diversity of microbial cultures is desired.

**Combination of culture-dependent and culture-independent methods.** Several studies have shown that culture-dependent and culture-independent methods often deliver different results, where each approach has its own biases and limitations (121, 122). Combined molecular and traditional approaches can, however, optimize the recovery of bacterial cultures (123, 124). Microbial communities and genomes (including those from the microbial dark matter fraction) can be, for example, characterized at a functional level using culture-independent approaches and bioinformatic tools (125–127). Such information can inform the development of specific culturing conditions to recover yet-to-be-cultured groups. The use of molecular tools to elucidate isolation patterns and improve the enrichment and cultivation processes was demonstrated for the bacteria *Campylobacter jejuni*, targeted from chicken feces (128).

Additionally, Sweet et al. (28) suggested the formulation of culture media with taurine to cultivate and recover obligate microbial symbionts from coral hosts. The authors observed enrichment of TauD (the gene for taurine dioxygenase) in genomes of CAMs, mainly in Oceanospirillales (28). TauD is related to the assimilation of host-derived taurine, a sulfur compound that is widespread in animal tissues and can be a marker gene of obligate symbionts in marine hosts such as coral and sponges (28, 129).

## FUTURE PROSPECTS

**Uncovering the hidden diversity of the coral microbiome.** To date, as discussed above, attempts to increase the culturable diversity of CAMs have had limited success, but the adaptation of some of the mentioned methods could overcome this. Diffusion-based approaches have been used to culture marine sponge-associated microbes (37, 43) through incisions made in the sponge tissue where the cultivation chambers (43) and iChips (37) devices are inserted into the holobiont. Although these incisions would not be as straightforward in corals (due to potential rejection, difficulties of inserting the devices into the skeleton, and the slow growth and regeneration of coral tissue), the devices could be placed near the coral colony in contact with the coral mucus, for example, without deleterious effects on the host. In addition, some adaptations to the manufacturing of the diffusion-based devices should be considered, in terms of the device composition and size. Large devices can disrupt the water flow surrounding corals and impact the incidence of light, consequently causing dysbiosis. Moreover, some materials (e.g., metals) can be toxic for corals and their symbionts. The use of inert polymers (for example polytetrafluoroethylene) would therefore be recommended.

We also propose the use of a coral-based culture medium for cultivation of CAMs (Fig. 1). The coral juice culture medium could be composed, for example, of ground coral nubbins (5 g) and then resuspended in 45 mL of saline solution at 2.5% wt/vol (to mimic the average salinity of seawater) and the addition of 5 mm glass beads. Suspensions would be incubated overnight in an orbital shaker at 120 rpm at room temperature. After incubation, this suspension would be centrifuged at 10,000 × *g* for 5 min and then the supernatant would be collected. Use of an autoclave is not recommended due to the composition of the mucus, which is rich in sugars that can caramelize during this process. Therefore, successive filtration steps, from 1- down to 0.22-$\mu$m pore size, would be required for purity. The filtered supernatant, the coral juice, should be added to autoclaved Marine Minimal Medium (see formulation described by Solano et al. [130]) at around 70°C. We recommend the use of approximately 5% of coral juice in 1,000 mL of media, followed by mixing the blend and plating 20 mL in each Petri dish. Coral juice will be the sole carbon source and a small concentration of glucose could be added. To ensure sterility, we recommend keeping the plated Petri dishes with coral juice culture medium at room temperature for at least 24 h before use.

Another example of a nature-based solution to improve the culturability of CAMs is to use the chemical composition of the coral mucus to selectively favor the growth of mucus-associated microbes (Fig. 2). Arabinose, fucose, mannose, C6 sugars, glucose, *N*-acetylglucosamine (GalNAc), xylose, galactose, and rhamnose are examples of components of coral mucus that appear to be species specific (67, 131, 132) and drive the

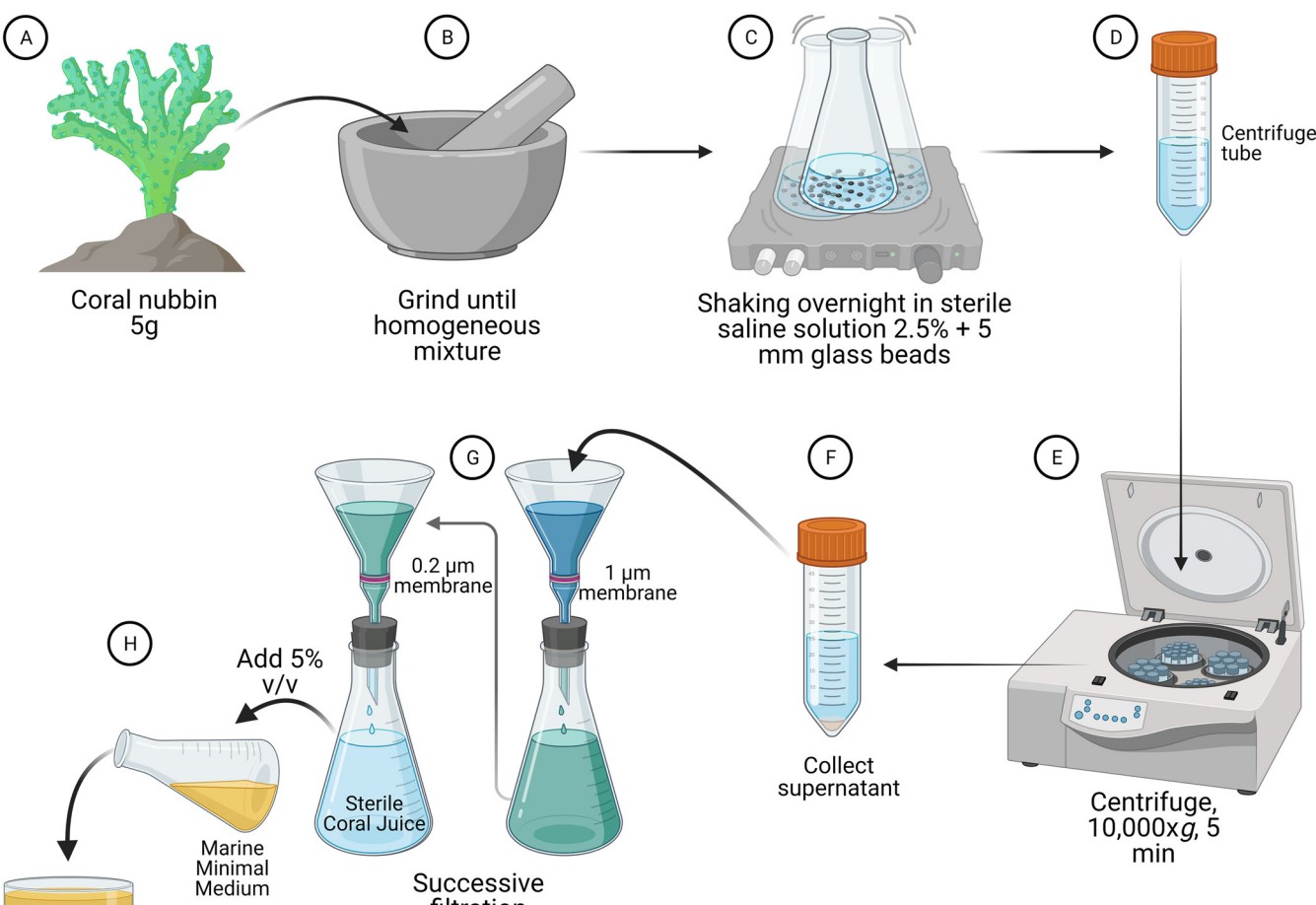

**FIG 1** "Coral juice" preparation for cultivation of coral-associated microorganisms. First, a 5-g coral fragment of interest (A) is macerated (B) to obtain a homogeneous mixture. Then, the mixture (C) is then added to an Erlenmeyer flask with 45 mL of saline solution (2.5%) and 5-mm glass beads for overnight incubation. After incubation, the contents are transferred into a centrifuge tube (D), centrifuged at 10,000 g for 5 min (E), and the supernatant is collected (F). Subsequently, successive filtration steps must be performed (G), starting with membranes of 1 μm (for debris retention) and ending with 0.22-μm pore size (for sterilization). The sterile supernatant is the coral juice. Then (H) 5% vol/vol of coral juice is loaded into autoclaved Marine Minimal Medium at 70°C, followed by mixing the blend and plating 20 mL in each Petri dish. Coral juice is the sole carbon source in the medium, favoring the growth of coral-associated microbes. After the seeding of samples of interest, plate dishes can be incubated under different oxygen demand conditions, for different periods of incubation and temperatures. Created using Biorender.com.

microbial composition of the host-associated microbiome. In corals, GalNAc and glucose changes seem to be related to the increase of Alphaproteobacteria, fucose and mannose are associated with decreases of Gammaproteobacteria, and arabinose and xylose can upregulate Cyanobacteria populations (67). Therefore, specific sugars could be used to compose selective media to isolate CAMs.

**The unacknowledged importance of coral-associated fungi.** Although over a decade of research has focused on the coral microbiome, efforts tend to favor coral-associated bacteria and the microalgal Symbiodiniaceae. There are fundamental gaps related to the diversity and potential functions of microeukaryotes (e.g., fungi and protists) in coral reef microbiomes (133, 134). The symbiotic relationships between coral and fungi, as well as fungal identity and diversity, are poorly understood and frequently accessed through culture-independent tools (134, 135). Here, we highlight the studies covering cultured coral-associated fungi and uphold the necessity to deeply explore the culturable fraction of this taxon.

To date, the known coral-associated fungi include members of the Ascomycota (dominant phylum), Basidiomycota, Mucoromycota, and Chytridiomycota phyla, with *Aspergillus* and *Penicillium* being the most frequently found genera (134, 136–138). Fungal associations with the coral species *Palythoa caribaeorum*, *Zoanthus sociatus*, *Palythoa variabilis*, and *Favia gravida* from shallow waters of southern Brazil were investigated by Paulino et

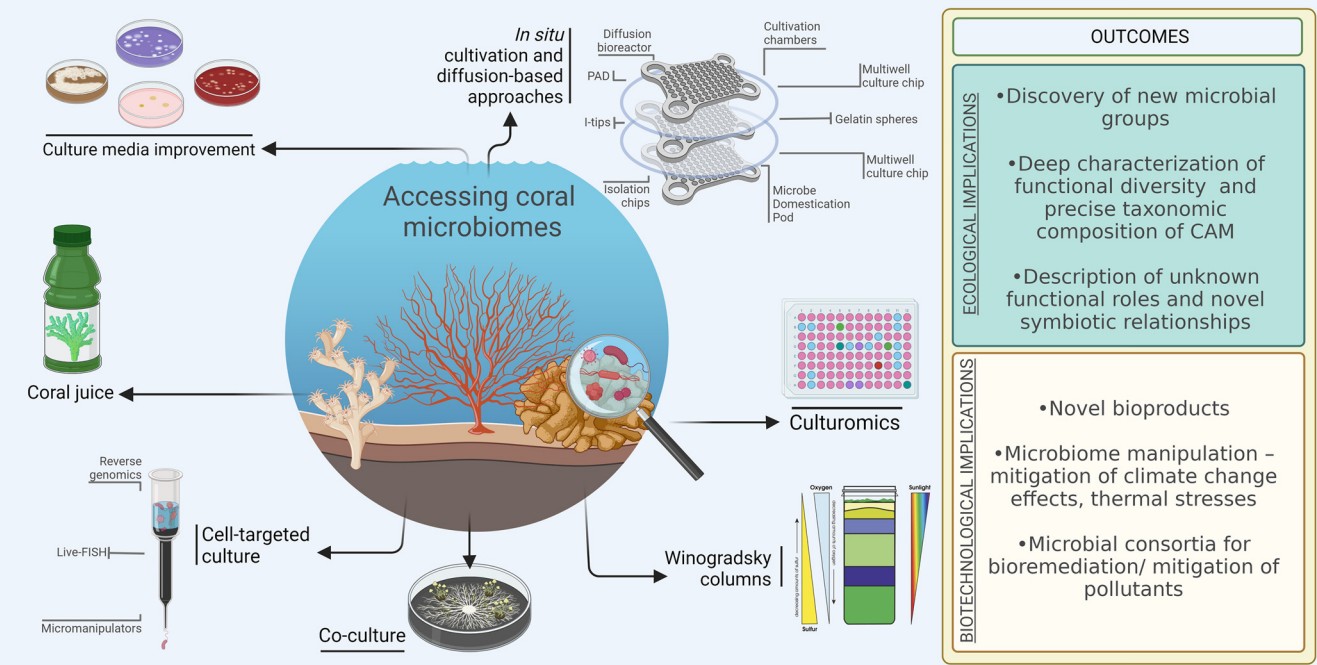

**FIG 2** Overview of the current knowledge of culture-dependent methods being used in coral microbiology, in addition to novel and alternative strategies that can be applied for culturing coral microbiomes. Created using Biorender.com.

al. (137); 50 strains were isolated and molecularly identified as *Aspergillus*, *Candida*, *Cladosporium*, *Clonostachys*, *Cordyceps*, *Hyphopichia*, *Microsphaeropsis*, *Neopestalotiopsis*, *Penicillium*, *Pestalotiopsis*, *Phoma*, *Pyrenochaetopsis*, *Talaromyces*, *Trichoderma*, and *Xylaria*. Similarly, these genera were also isolated from coral specimens collected in the Red Sea (136). The culture media used for isolation were dextrose yeast extract agar, rose bengal agar (136), and seawater malt extract agar supplemented with chloramphenicol (137). Both studies reported rapid growth genera; the assessment of slow-growing or hard-to-culture fungi is still necessary to ensure a broader view of the diversity of cultured fungi. Focusing beneath the coral surface, endolithic fungi (e.g., representatives of Ascomycota and Basidiomycota) were found to penetrate the calcium carbonate microstructures and interact with the coral cells (138–140).

The cultured fraction of coral fungi remains underexplored, and existing studies are mostly related to coral diseases, focused on either parasitic or opportunistic interactions. Several studies have identified pathogenic fungi (140–143), in which *Aspergillus sydowii* is the most studied and characterized fungus present in healthy and diseased corals and is involved in the emerging fungal disease Aspergillosis, which drastically affects coral reefs (139, 144). On the other hand, evidence indicates that fungi may be related to complex metabolic activities in coral holobionts, as well as modulating coral health and resilience (135, 142, 145, 146).

In addition to the relevance of fungal pathogens in coral health, fungi are an important player in the symbiotic relationship with the metaorganism, capable of driving key steps of the biomineralization of the coral skeleton, in biological nitrogen and carbon cycling, as well as providing antimicrobials and antioxidant substances or inducing the host to produce them (136, 139, 145, 146). It is known that antimicrobial substances play a crucial role in the modulation of the coral microbiome, preventing the settlement and proliferation of potential pathogens (4, 139, 140). Antiradical antioxidants are important for the maintenance of host-Symbiodiniaceae relationships under coral stress (4). Coral fungi can also produce protective molecules, such as mycosporine-like amino acids (147), and can enhance the survival of skeletogenic cell types against UV irradiation (148).

Coral-associated fungi are also important producers of bioproducts with many biotechnological implications. For example, they can synthesize polyketides with antibacterial and antifouling activity (149, 150), different bioactive peptides (151, 152), terpenoids (153, 154), and even neuronal modulators (153, 155), among other secondary metabolites. Furthermore, coral fungal diversity can be utilized in water bioremediation because of its ability to degrade contaminants. Filamentous fungi (species belonging to the genera *Aspergillus*, *Cladosporium*, *Penicillium*, and *Trichoderma*) isolated from *Mussismilia hispida*, *Palythoa caribaeorum*, *Palythoa variabilis*, and *Zoanthus solanderi* were able to efficiently degrade Remazol Brilliant Blue R (textile dye) (156). Additionally, a multidomain microbial consortium (including cultured the fungi *Geotrichum* sp., *Rhodotorula mucilaginosa*, and *Penicillium citrinum*) was developed as a tool in bioremediation of oil on a mesocosm scale, in which the consortium significantly helped to mitigate the impacts of crude oil, substantially degrading the polycyclic aromatic and n-alkane fractions and maintaining the physiological integrity of the hydrocoral *Millepora alcicornis* (10).

Based on the putative role of coral fungi in the symbiotic relationship and great potential of exploiting fungal diversity for biotechnology, we suggest that studies on the cultivation of coral-associated fungi be expanded. The goal would be to populate culture collections and shed light on the phylogenetics, lifestyle, genomics, and biotechnological potential of the symbiotic relationship between the fungal community and the coral holobiont.

## FINAL REMARKS

Culturing novel microbial groups associated with corals can contribute to coral reef preservation efforts, bioprospecting of novel microorganisms with biotechnological potential, and improved insight into the biology and ecology of the coral holobiont. Data from the Coral Microbiome Database show that 87.4% of coral-associated bacteria and archaea sequences are related to uncultured microbes and only 6.5% to cultured microbes (27).

A timely example, considering the urgent need for tools that can extend and accelerate the capacity of corals to adapt to climate change (13, 157), is the use of beneficial microorganisms for corals (BMCs) as probiotics to promote coral health (3, 4). A few studies have already demonstrated the efficacy of BMCs and microbiome manipulation to promote health in different coral species, such as *Pocillopora damicornis* (7, 11), *Mussismilia hispida* (9), *Mussismilia hartii* (158), *Millepora alcicornis* (10), *Acropora millepora* (8), *Pocillopora* sp., and *Porites* sp (159). Although the application of coral slurry can promote coral health experimentally (159), the use of microbial isolates, or BMCs, represents the most feasible "customized medicine for corals" currently being developed (32), and culturing coral microbial dark matter would greatly contribute to the selection of more potent BMCs. In addition, secondary metabolites (i.e., antimicrobial, antifouling, antitumor, antiparasitic, antiviral compounds, and biosurfactants) are naturally produced by CAMs to prevent growth of competitors and/or deter coral pathogens/disease (160–163). Therefore, they represent interesting targets for other biotechnological applications and pharmaceutical resources.

It is also important to acknowledge that the group of currently uncultured bacteria covers dominant and ubiquitous taxa, resulting in knowledge gaps regarding some symbiotic microbes of corals. *Endozoicomonas*, for example, is a ubiquitous genus widely found in corals from the deep sea to shallow waters (15, 164). This bacterial group maintains symbiotic relationships with corals, and genomic evidence shows that they contribute to DMSP metabolism (165), vitamin B provisioning (166), carbohydrate cycling, and conversion of nitrate to nitrite (157), all of which are key functions for the host. However, there are just a few cultured *Endozoicomonas* sp. representatives described, due to the difficulty in cultivating this genus (45, 165). Thus, most of their capabilities and functions for the host remain unknown. Overall, culturing key microbial groups is therefore a first and most crucial step to identify and, subsequently, manipulate their symbiotic roles for corals.

## ACKNOWLEDGMENTS

This research paper was carried out in association with the ongoing R&D project registered as ANP 21005-4, "PROBIO-DEEP–Survey of potential impacts caused by oil and gas exploration on deep-sea marine holobionts and selection of potential bioindicators and bioremediation processes for these ecosystems" (UFRJ/Shell Brasil/ANP), sponsored by Shell Brasil under the ANP R&D levy as "Compromisso de Investimentos com Pesquisa e Desenvolvimento." R.S.P. was supported through KAUST grant number BAS/1/1095-01-01 and KAUST Center Competitive Funding (CCF) FCC/1/1973-51-01). F.M. received support from the National Council for Scientific and Technological Development (CNPq) and the National Council for the Improvement of Higher Education (CAPES).

We declare that the research was conducted in the absence of any commercial or financial relationships that could be construed as a potential conflict of interest.

All authors listed have made substantial, direct, and intellectual contributions to the work and approved it for publication.

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
