## [Reviewer comments · mSystems]

Methods and strategies to uncover coral-associated microbial dark matter

Raquel Peixoto, Junia Schultz, Fluvio Modolon, Alexandre Rosado, Christian Voolstra, and Michael Sweet

Corresponding Author(s): Raquel Peixoto, King Abdullah University of Science and Technology

Review Timeline:

Submission Date:	April 18, 2022
Editorial Decision:	May 30, 2022
Revision Received:	June 9, 2022
Accepted:	June 14, 2022

Editor: Neha Garg

Reviewer(s): The reviewers have opted to remain anonymous.

Transaction Report:

DOI: <https://doi.org/10.1128/msystems.00367-22>

May 30, 2022

Prof. Raquel Peixoto
King Abdullah University of Science and Technology
Thuwal
Saudi Arabia

Re: mSystems00367-22 (Methods and strategies to uncover coral-associated microbial dark matter)

Dear Prof. Raquel Peixoto:

Thank you for submitting your manuscript to mSystems. We have completed our review and I am pleased to inform you that, in principle, we expect to accept it for publication in mSystems. However, acceptance will not be final until you have adequately addressed the reviewer comments.

Preparing Revision Guidelines

Sincerely,

Neha Garg

Editor, mSystems

Journals Department
Reviewer comments:

Reviewer #1 (Comments for the Author):

Schultz et al. have reviewed various aspects about culturing microbes, specifically from corals. Overall, I liked the review and the strengths are that it is collecting and centralizing data on culturing coral-associated microbes (CAMs). The main weakness I see in this paper is that for some aspects it is very surface level and there's not enough depth to the some sections to really convey the complexity of culturing microbes. I am in no way saying that anything is wrong here, I'm mostly suggesting that some sections could have more depth and topics also be covered.

General comments:

1) Can the authors ensure they are using the already established terminology for certain things. I have more context in my specific comments.

2) There's no in depth/specific mention of enrichment of close culturing systems and the accumulation of toxic byproducts (apologizes if I missed this). There is some, but I'm referring to when the authors refer to simply adding sugars or chemicals like DMSP to media. There should be a caveat added of toxic byproducts accumulating from the metabolism of these additives. For example, it is known that glycerol present in growth media with certain *Vibrio* sp. will result in the accumulation of acidic byproducts that cause rapid culture crashes.

3) Maybe to add context, it would be good to add the relative year when these techniques were developed. For example, culture medias have not really changed in decades so it worth noting this is a major reason they should be updated.

Specific comments:

Line 53-54: I would include the endosymbionts as well in this statement.

Line 139 (entire paragraph): Can there be more of a comparison of these techniques? Maybe pros and cons? I appreciate the overview, but the content/description is sparse here. I think this section can benefit from more elaboration.

Line 147: The authors are missing aspects about the toxic nature of lysed coral cells. What about the toxic montiporic acids released from *Montipora* spp.?

Line 183: The correct term is lysogeny broth. However, there are also modifications used instead, like LB + salt (LBS) that are actually being used for marine bacteria.

Line 184: This is more typically referred to as GASW Agar instead of GASWA. But the are different formulations and modification to this media as well (please note that).

Line 186: Is it actually NA or did the authors modifying? If is was modified, it can no longer be called NA.

Line 187: I'm not comfortable with this statement. First, there is a comparison of selective media vs general media. Second, this is from a literature review paper and the actual medias are not being directly compared. I this this section is a bit misleading.

Line 189 (paragraph): Can the authors please use the standard terms for media types?

Basal/general media, enriched media, selective media, indicator media, transport media, and storage media.

Line 192-193: The wording is technically incorrect. This should be worded that you will increase the diversity of bacteria cultured. You increase the number of bacterial cells from growing them.

Line 197: "Promoters" This means something very different for genetics. Please use a different term. Or just say enrichment.

Line 201 (paragraph): The concept of nutrient shock should be mentioned. This is not a new phenomenon. This is was lead to the development of R2A.

Line 205: Again, I agree this was "new" for coral research, but these are known microbiology techniques and the original research should at least be acknowledged.

Table: I like this table, but I think it would benefit from the relative year these technologies were developed. I'm always worried people will immediately disregard modification of culture medias and only focus on the fancy new toys available. But to give

some context that culture medias are outdated and need improvement might make people realize that "simple" modifications can be combined with technology.

I would like to say that I enjoyed reading this manuscript and I wish you all the best in the revision process.

June 9, 2022

Dear editor,

We are sending attached the revised version of the manuscript “Methods and strategies to uncover coral-associated microbial dark matter”, and the associated point-by point answers to the reviewer, where we address all the suggestions and comments provided.

We would like to thank you and the reviewer for the detailed review and constructive suggestions provided.

Reviewer #1 (Comments for the Author):

Schultz et al. have reviewed various aspects about culturing microbes, specifically from corals. Overall, I liked the review and the strengths are that it is collecting and centralizing data on culturing coral-associated microbes (CAMs). The main weakness I see in this paper is that for some aspects it is very surface level and there's not enough depth to the some sections to really convey the complexity of culturing microbes. I am in no way saying that anything is wrong here, I'm mostly suggesting that some sections could have more depth and topics also be covered.

A: Thank you very much for the positive feedback and suggestions to improve our manuscript.

General comments:

1) Can the authors ensure they are using the already established terminology for certain things. I have more context in my specific comments.

A: Thank you. We have double checked the terminology used, as detailed below.

2) There's no in depth/specific mention of enrichment of close culturing systems and the accumulation of toxic byproducts (apologizes if I missed this). There is some, but I'm referring to when the authors refer to simply adding sugars or chemicals like DMSP to media. There should be a caveat added of toxic byproducts accumulating from the metabolism of these additives. For example, it is known that glycerol present in growth media with certain *Vibrio* sp. will result in the accumulation of acidic byproducts that cause rapid culture crashes.

A: Thank you for the note. We added more information regarding these points in the third paragraph of topic 2.2.1 “Culture media and growth conditions: regular and modified media”, as follows:

“Also, the selective DMSP-enriched culture medium seems to be a good option to increase the diversity of obtained CAMs [29], which is an example that, despite the potential accumulation of selective byproducts from the metabolism of culture media additives, such as sugars and or chemical compounds like DMSP [68], the selection of the media composition should be based on the scientific goals.” We also mention other toxic compounds and conditions in lines 125-129; 161-165; 251-262, in Table 1 and in our conclusions.

3) Maybe to add context, it would be good to add the relative year when these techniques were developed. For example, culture medias have not really changed in decades so it worth noting this is a major reason they should be updated.

A: Thank you for the suggestion. We added this information in the revised manuscript, specifically in the first paragraph of topic 2.2.1. “Culture media and growth conditions: regular and modified media” and 2.2.3 “Alternative approaches”, as follows:

“Many resources are required for a microbe to be cultured, such as the correct physicochemical nutrients and energy sources. The most basic approach used to culture microbes from different sources is the use of nutrient-rich culture media (e.g., tryptic soy broth, lysogeny broth, marine broth, nutrient broth, to name a few). These culture media were developed decades ago (more than 50 years), and have been applied to grow and maintain microbial cultures from different environments and hosts, including the coral holobiont, using relatively few modifications that can increase the microbial diversity obtained. Here, we highlight that novel approaches developed for other hosts and samples should be adapted to increase CAM culturabilty.”

Specific comments:

Line 53-54: I would include the endosymbionts as well in this statement.

A: Good point. The endosymbionts are already included, as part of the microbiome, but it might be indeed good to highlight it. This sentence has been therefore edited, as follows: “Beneficial microbes are essential members of the coral holobiont (i.e., the coral host plus the associated microbiome, including their endosymbionts) [1, 2].

Line 139 (entire paragraph): Can there be more of a comparison of these techniques? Maybe pros and cons? I appreciate the overview, but the content/description is sparse here. I think this section can benefit from more elaboration.

A: We appreciated the suggestion and highlight that our goal here was indeed to be descriptive, as a comparison of these approaches would likely be somehow biased, due to the fact that these protocols were never directly compared, at least not all of them. We agree thought that some context/discussion on this direction could be provided, so we edited this paragraph as follows:

“Microbes associated with mucus, for example, can be collected using sterile syringes to extract the liquid from the coral surfaces, or even from the whitish slurries that can form around the corals [53, 54]. Alternatively, corals can be placed upside down in sterile flasks without water, from which mucus secretion can be retrieved after 20-30 min, sometimes referred to as coral ‘milking’ [55]. Also, a commonly used technique for sampling mucus is swabbing the coral surface [56, 57]. Additionally, the mucus can be sampled by low-speed centrifugation [58, 59]. Finally, the aptly named ‘snot sucker’, an apparatus that allows for the separation of the two distinct layers of the surface mucus, can also be used [60]. Each of these protocols present associated challenges and limitations, including the amount of biomass that is obtained, potential microbial contamination from the surrounding seawater and/or cross-contamination among coral compartments. The selected technique will therefore likely rely on the available expertise, logistics and research goals.”

Line 147: The authors are missing aspects about the toxic nature of lysed coral cells. What about the toxic montiporic acids released from *Montipora* spp.?

A: Very interesting point we were not aware of, thank you for including it. This information is now available in the third paragraph of the topic 2.1 “Sample processing interferes with the culturability of the coral microbiome”. Check the new sentences below:

“It is important to mention the potential effect of coral holobiont compounds on the culturability of specific CAMs. The production of mantiporic acids by the stony coral *Montipora* spp., [63, 64], can, for example, affect cultured cells (e.g., zooxanthellae), due to its antimicrobial activity and cytotoxicity [64, 65].”

Line 183: The correct term is lysogeny broth. However, there are also modifications used instead, like LB + salt (LBS) that are actually being used for marine bacteria.

A: Thank you for the note. It was a typo and we corrected it in the revised manuscript. The term “lysogen broth” was corrected to “lysogeny broth”

Line 184: This is more typically referred to as GASW Agar instead of GASWA. But the are different formulations and modification to this media as well (please note that).

A: We checked the culture media composition and as they are the same (GASWA and GASW Agar), we replaced it by GASWAgar in the manuscript.

Line 186: Is it actually NA or did the authors modifying? If is was modified, it can no longer be called NA.

A: We have double checked and it was not a modified nutrient agar. In this way, we kept calling NA.

Line 187: I'm not comfortable with this statement. First, there is a comparison of selective media

vs general media. Second, this is from a literature review paper and the actual medias are not being directly compared. I think this section is a bit misleading.

A: We agree the use of these media were not compared for the same samples and conditions, although we meant the overall reported diversity obtained by each media. We agree that these aspects can be misleading if not clearly presented, and edited the entire 2.2.1 Culture media and growth conditions: regular and modified media section, accordingly.

Line 189 (paragraph): Can the authors please use the standard terms for media types? Basal/general media, enriched media, selective media, indicator media, transport media, and storage media.

A: This paragraph (and section) has been reviewed for the use of the standard terms “2.2.1 Culture media and growth conditions: regular and modified media”.

Line 192-193: The wording is technically incorrect. This should be worded that you will increase the diversity of bacteria cultured. You increase the number of bacterial cells from growing them.

A: Good catch. Thank you. This has been edited to:

“Thus, combining different culture media and applying variable incubation conditions will undoubtedly increase the diversity of cultured bacteria.”

Line 197: "Promoters" This means something very different for genetics. Please use a different term. Or just say enrichment.

A: The term “promoters” was removed from the text.

Line 201 (paragraph): The concept of nutrient shock should be mentioned. This is not a new phenomenon. This is what led to the development of R2A.

A: We highlighted the nutrient shock phenomenon as follows:

“The excessive amount of nutrients from rich culture media can cause a high nutrient shock and impair the growth of microorganisms from stressful low-nutrient environments [71]. Keller-Costa et al. [70] isolated a plethora of strains associated with the soft coral *Eunicella labiata* using a basic modification in marine agar, which was diluted (1:2) in artificial seawater, aiming to deplete part of the carbon source. With this relatively simple modification, 416 morphologically distinct colonies were obtained, corresponding to 62% of the bacterial phylotypes associated with the gorgonian.”

Line 205: Again, I agree this was "new" for coral research, but these are known microbiology techniques and the original research should at least be acknowledged.

A: This is a very good point and we have added the original reference here: “In brief, sediment samples were enriched for six months using an anaerobic basal medium supplemented with 1.0 mg/L *Escherichia coli* genomic DNA at 28 °C, as previously described by Fardeau et al. [73]”, and in other parts of the text.

Table: I like this table, but I think it would benefit from the relative year these technologies were

developed. I'm always worried people will immediately disregard modification of culture medias and only focus on the fancy new toys available. But to give some context that culture medias are outdated and need improvement might make people realize that "simple" modifications can be combined with technology.

A: The year of each publication/protocol was included as part of the table and also throughout the text.

I would like to say that I enjoyed reading this manuscript and I wish you all the best in the revision process.

A: Thank you very much for the kind feedback and valuable suggestions that have significantly contributed for the improvement of the manuscript.

June 14, 2022

Prof. Raquel Peixoto
King Abdullah University of Science and Technology
Thuwal
Saudi Arabia

Re: mSystems00367-22R1 (Methods and strategies to uncover coral-associated microbial dark matter)

Dear Prof. Raquel Peixoto:

Your manuscript has been accepted, and I am forwarding it to the ASM Journals Department for publication. For your reference, ASM Journals' address is given below. Before it can be scheduled for publication, your manuscript will be checked by the mSystems production staff to make sure that all elements meet the technical requirements for publication. They will contact you if anything needs to be revised before copyediting and production can begin. Otherwise, you will be notified when your proofs are ready to be viewed.

Publication Fees:

We recognize that the video files can become quite large, and so to avoid quality loss ASM suggests sending the video file via <https://www.wetransfer.com/>. When you have a final version of the video and the still ready to share, please send it to mSystems staff at mssystems@asmusa.org.

For mSystems research articles, if you would like to submit an image for consideration as the Featured Image for an issue, please contact mSystems staff at mssystems@asmusa.org.

Sincerely,

Neha Garg

Editor, mSystems

Journals Department
E-mail: mSystems@asmusa.org